# Development Features of *Ixodes ricinus* × *I. persulcatus* Hybrids under Laboratory Conditions

**DOI:** 10.3390/microorganisms11092252

**Published:** 2023-09-07

**Authors:** Oxana A. Belova, Alexandra E. Polienko, Anastasia D. Averianova, Galina G. Karganova

**Affiliations:** 1Laboratory of Biology of Arboviruses, FSASI “Chumakov FSC R&D IBP RAS” (Institute of Poliomyelitis), 108819 Moscow, Russiakarganova@bk.ru (G.G.K.); 2Institute for Translational Medicine and Biotechnology, Sechenov University, 119991 Moscow, Russia

**Keywords:** *Ixodes ricinus*, *Ixodes persulcatus*, hybrids, life cycle, morphogenetic diapause

## Abstract

Widely distributed *Ixodes ricinus* and *Ixodes persulcatus* ticks transmit many pathogens of both medical and veterinary significance. The ranges of these tick species overlap and form large sympatric areas in the East European Plain and Baltic countries. It has previously been shown that crossing *I. ricinus* and *I. persulcatus* is possible, resulting in the appearance of sterile hybrids. In the present study, we analyzed the features of this hybrid’s life cycle under laboratory conditions. For this purpose, virgin females of *I. ricinus* and *I. persulcatus* ticks were obtained in the laboratory, and hybrid generations of ticks were bred from the reciprocal crossings of these two tick species. According to our data, mating the females of *I. ricinus* and *I. persulcatus* with the males of another species leads to a decrease in the engorgement success of the females, a decrease in the number of hatched larvae, and the appearance of a hybrid generation in which both females and males are sterile. Under laboratory conditions at a constant room temperature and under natural daylight, the morphogenetic diapause of the engorged *I. persulcatus* larvae began in September. For *I. persulcatus* nymphs, it occurred earlier than for *I. ricinus*, in October and November, respectively. The hybrids generally repeated the features of the life cycle of the mother species.

## 1. Introduction

*Ixodes ricinus* (L., 1758) and *Ixodes persulcatus* Sch., 1930 ticks (Acari, Ixodidae) transmit many pathogens of both medical and veterinary significance, including causative agents of tick-borne encephalitis (TBE) and Lyme borreliosis. These ticks are widely distributed: *I. ricinus* inhabits northern, western, central, and eastern Europe [1,2], and *I. persulcatus* has an extensive Eurasian range from the Baltic region to the Pacific [1,3]. The ranges of these tick species overlap and form large sympatric areas in the East European Plain and Baltic countries [4,5,6,7,8,9].

The ecology and biology of *I. ricinus* and *I. persulcatus* ticks have been well studied and reviewed in many works [2,10,11,12,13,14,15]. The developmental cycles of *I. ricinus* and *I. persulcatus* include four stages: egg, larva, nymph, and adult. These stages, except eggs, can exist in three phases: unfed, feeding, and engorged. The feeding duration is rather short: 3–6 days for immature phases and 6–12 days for adult females [10]. For the rest of the life cycle (approximately 98% of the time), individuals of different phases inhabit the environment under the influence of abiotic and biotic factors. Due to the wide range and different climatic conditions of the ticks’ habitats, the total durations of the developmental cycles of *I. ricinus* and *I. persulcatus* vary from 3 to 6 years [10,11,16,17]. Under the conditions of seasonal climatic rhythms, periods of tick activity are interrupted by periods of dormancy, which may be represented as diapause or quiescence [11,16,18]. Quiescence is controlled by the temperature and humidity of the environment and appears as an immediate response to unfavorable conditions, while diapause is a period of hormonally controlled arrested development. Diapause usually occurs seasonally before environmental conditions become unfavorable and is mostly controlled by the photoperiod [11,16,18]. Most authors distinguish two main types of tick diapause: morphogenetic (developmental) and behavioral. During morphogenetic diapause, there is a delay in the development of laid eggs and an elongation of the period of preparation for molting in engorged larvae and nymphs or oogenesis in females [16]. Behavioral diapause involves the temporary suppression of host seeking and/or attachment by unfed ticks, and this can appear immediately after the hatching or molting of ticks [16].

Seasonal tick activities, periods of induction, regulation, and types of diapause differ in *I. ricinus* and *I. persulcatus* not only between species but also between populations of the same species from different parts of the range [11,19]. In a temperate climate, larvae and nymphs of both species continue to be active throughout the warm season with more or less pronounced peaks in the spring and autumn [11,20]. Larvae and nymphs can overwinter both in unfed and engorged states and have an optional diapause corresponding to a long-day photoperiodic reaction. It has been proven that engorged immature phases have photoperiodic sensitivity throughout the entire period of motility, and the development of morphogenetic diapause is determined by the conditions before and after tick feeding (a two-step photoperiodic reaction of the short-day/long-day type) [11,18]. The seasonal activity of adult *I. ricinus* and *I. persulcatus* differs greatly. In a temperate climate, the activity of adult *I. ricinus* is confined to the spring–summer–autumn period; in summer, there may be a decrease in tick activity depending on climatic conditions [10,11]. Due to facultative behavioral diapause, adult *I. ricinus* can become active during the nymph molting season and take part in the autumn peak of activity [11]. The activity of adult *I. persulcatus* is observed in the spring immediately after the snow melts and until the middle of summer. In contrast with *I. ricinus*, behavioral diapause is obligate in unfed adult *I. persulcatus*, which prevents the autumn activity of ticks [11,12]. Another difference is that engorged *I. persulcatus* females do not have diapause and are not capable of overwintering. The differences between these species also concern the phase of the egg. Facultative embryonic diapause is characteristic of *I. ricinus*, while *I. persulcatus* eggs always develop without diapause and are not capable of overwintering [10,11]. These notable differences are interpreted as an adaptation of *I. persulcatus* to harsher climatic conditions and shorter warm seasons in the taiga region [11,16].

Under the current conditions of climate change and intense anthropogenic impacts, the ranges of *I. ricinus* and *I. persulcatus* and their zones of sympatry are changing, which in some cases has led to changes in the eco-epidemiology of tick-borne diseases [17,21,22,23,24,25,26,27,28,29]. It should be noted that the infection rates of *I. ricinus* and *I. persulcatus* with some pathogens in the allopatric and sympatric territories can be different: TBEV and *Borrelia burgdorferi* sensu lato prevalence was usually higher in *I. persulcatus* than in *I. ricinus*, both in the zone of dominance and in the zone of sympatry [5,6,7,30,31,32]; at the same time, *Anaplasma phagocytophilum* seroprevalence in dogs in Latvia was significantly higher in *I. ricinus*-dominant and sympatric territories than in *I. persulcatus*-dominant areas [33]. Furthermore, in sympatry zones, *I. ricinus* and *I. persulcatus* ticks might share the same hosts and habitats, and the exchange of pathogens is possible [7,9,30,34], which in turn can affect the properties of the pathogens.

Previously, it was shown that within the zones of *I. ricinus* and *I. persulcatus* sympatry, the existence of hybrid forms of these tick species is possible [35,36,37]. For the first time, the possibility of the existence of first-generation hybrids was proved in laboratory experiments with the reciprocal crossing of *I. ricinus* and *I. persulcatus* ticks in the 1990s when the sterility of the obtained hybrids was also noted [20]. Subsequently, we were able to repeat the experiments and obtain the hybrids of the indicated species of ticks in order to study the morphology of the hybrids and to find hybrid individuals in nature [36,37]. Although the ITS2 region is not a reliable marker to distinguish real hybrids [38], the existence of hybrid ticks in Estonia [35] and Finland [27] has been noted.

Experiments with mosquitoes showed that hybridization can impact vector competence [39], which in turn affects the risk of infection in natural foci. Recently, we were able to prove the competence of hybrids as TBEV vectors [40]. We showed that the hybrids from crossing *I. persulcatus* females with *I. ricinus* males were able to effectively acquire and transmit the TBEV of Siberian and European subtypes trans-stadially and to a susceptible host. Also, in hybrids, we observed the highest acquisition effectiveness and RNA copy numbers during Siberian TBEV subtype transmission [40]. To assess the local risk of infection with a tick-borne pathogen, it is necessary to have information about not only the tick infection rate and the properties of the pathogen but also about the seasonal course of activity and features of the life cycles of ticks. While the life cycles of ‘pure’ *I. ricinus* and *I. persulcatus* have been studied well enough, the features of the life cycles of hybrids have not yet been described. In this study, we analyzed long-term data on the development of hybrids of *I. ricinus* and *I. persulcatus* under laboratory conditions in comparison with the original species.

## 2. Materials and Methods

### 2.1. Ticks

Ticks were collected via flagging in allopatric territories in Russia: *I. ricinus* in the Voronezh (51°37′48″ N, 39°40′17″ E) and Kaliningrad (55°09′32″ N, 20°50′37″ E) regions, and *I. persulcatus* in the Republic of Karelia (62°04′05″ N, 33°58′17″ E). From these ticks, laboratory colonies of *I. ricinus* and *I. persulcatus* ticks were obtained and reared through one or two generations under controlled laboratory conditions. In the laboratory, living ticks were kept in humidified glass tubes that have been described earlier [41]. To maintain the laboratory cultures, adult ticks were fed on rabbits (*Oryctolagus cuniculus* (L., 1758), breed “Soviet Chinchilla”, “Scientific Center of Biomedical Technology” RAS, branch “Stolbovaya”) under a cloth cap, as described earlier [42]. Larvae and nymphs were fed on laboratory mice (*Mus musculus* L., 1758, outbred ICR mice, “Scientific Center of Biomedical Technology” RAS, branch “Stolbovaya”), as described earlier [40]. All animals were maintained according to the international guidelines for animal husbandry, including the recommendations of CIOMS, 1985, and the FELASA Working Group Report, 1996–1997. The study protocol was approved by the Ethics Committee of Chumakov FSC R&D IBP RAS (Institute of Poliomyelitis) (protocol no. 040321-1 from 4 March 2021).

To breed hybrids, in the first stage, virgin *I. persulcatus* and *I. ricinus* females were obtained (Figure 1a). The sex of ticks cultured in the laboratory was determined at the engorged nymph stage [43,44]. Large and small engorged nymphs were separated into different tubes, where they molted into males and females. In the second stage, virgin females were placed in glass tubes with males of other species for several days, and then they were fed on rabbits (Figure 1b). To obtain adult hybrid ticks, immature hybrids were fed on laboratory mice. To avoid mating, female and male hybrids were separated into different glass tubes at the engorged nymph stage. Several *I. ricinus* and *I. persulcatus* ticks and obtained hybrids were analyzed genetically to confirm the ‘purity’ of the original species and the successful breeding of a hybrid generation [38]. The obtained hybrid adult ticks were used in reciprocal crossings with the ticks of the original species or with hybrids.

### 2.2. Study of the Life Cycles of Ticks under Laboratory Conditions

Analyses of the life cycles of *I. ricinus* and *I. persulcatus* and their hybrids under laboratory conditions were carried out using data accumulated during the maintenance of the laboratory cultures of the ticks. The ticks were kept at room temperature (20–23 °C) and under natural light in glass tubes throughout their entire life cycle. The biological age of all ticks before feeding was 30–90 days (after hatching/molting). During the feeding of adult ticks, for each female, the duration of feeding, the weight, the starting date of oviposition, and the starting date of larvae hatching were recorded. The success of females’ feeding was also recorded. Females that detached from the host on their own after feeding were considered successfully engorged. During the feeding of immature ticks, for a group of 100–150 larvae or 15–20 nymphs fed on one mouse, the duration of feeding and the starting date of molting were recorded.

### 2.3. Statistical Analysis

Statistical analyses were performed using OriginPro 8 SR4 (v8.0951, Northampton, MA, USA). To compare various rates between groups (oviposition, hatching, molting, etc.), Fisher’s exact test or the chi-square test was used. For other data, the significance of differences between groups was estimated according to the Mann–Whitney U test (for 2 groups) and the Kruskal–Wallis H test (for 3 or more groups). For multiple comparisons, Bonferroni correction was applied.

## 3. Results

### 3.1. Feeding and Oviposition of Female Ticks Depending on the Species of the Mating Partner

In this experiment, the main parameters of female engorgement, oviposition, and the hatching of larvae were assessed after the copulation of females with males from different groups (Figure 2, Table 1 and Appendix A). Many of these parameters were significantly affected by the feeding season of females (Figure 2, Appendix A). Thus, the maximum weight of the engorged females of both species was observed after feeding in autumn, and the minimum was observed in the summer (Figure 2a, Appendix A). Other indicators also depended on the season of female feeding: in spring, a shorter duration of female feeding was observed compared with the summer period (the differences were significant for all groups, except for pure *I. persulcatus*; Figure 2b); for females fed in summer, the shortest period from engorgement to the beginning of oviposition was noted (significant for all groups, for female and male *I. ricinus*–*I. persulcatus* pairs); in pure *I. ricinus* and in the female and male *I. persulcatus*–*I. ricinus* pairs, the longest period from the beginning of oviposition to the beginning of larvae hatching was observed in autumn (the differences were significant when compared with the summer; Figure 2c).

To compare the characteristics of the feeding and oviposition of *I. ricinus* and *I. persulcatus*, and hybrid females depending on the species of the mating partner, we analyzed the combined data (Table 1) since intergroup differences were observed regardless of the feeding month of the females.

*Ixodes persulcatus* females, after mating with males of their own species, showed significantly higher engorgement success (the proportion of females that independently detached from the host after feeding) compared with females that mated with *I. ricinus* males (chi-square test, *p* < 0.01; Table 1). For *I. ricinus*, such differences were not significant. It should be noted that the proportion of engorged *I. ricinus* females that mated with *I. persulcatus* males was significantly higher than in the reverse situation (chi-square test, *p* < 0.05; Table 1). Mating females with the males of another species also resulted in a decrease in larvae hatching success (Table 1). The substantial and synchronous hatching of larvae was observed only in the clutches of pure species. In the clutches of females that mated with a male of another species, the hatching of larvae was always long and often did not exceed 50%.

The average weight of engorged *I. persulcatus* females was higher than that of *I. ricinus* after crossing with a male of both their own species and another species (Mann–Whitney test, *p* < 0.05; Table 1). When females of the original species were crossed with hybrid males, the average weight of the engorged females was slightly lower than that of the original species, but the difference was not significant.

The mating of the original species with hybrid males and the mating of hybrid females with males, both of the original species and hybrids, confirmed the previously stated observation about the sterility of hybrids. Despite the fact that the engorgement success of females from the mentioned crosses averaged 44.1% and the proportion of females that laid eggs averaged 61.5%, larvae from such clutches did not hatch at all, or only 1–2 individuals hatched (Table 1).

Thus, the weight of engorged females and the duration of feeding, oviposition, and hatching of larvae depended on the feeding season of the females. The mating of *I. ricinus* and *I. persulcatus* females with the males of another species led to a decrease in the engorgement success of the females (significant for *I. persulcatus*), a decrease in the larvae hatching success, and the appearance of sterile hybrid offspring. The females and males of the hybrids were both sterile (with a few exceptions).

### 3.2. Development of I. ricinus, I. persulcatus, and Hybrid Larvae and Nymphs under Laboratory Conditions Depending on the Feeding Season

In this experiment, the duration of the feeding of immature ticks and the date of the start of molting were recorded for groups of 100–150 larvae or 15–20 nymphs, which fed on one mouse. The results are presented in Figure 3, Appendix A.

The feeding durations of *I. ricinus* and *I. persulcatus* larvae differed significantly (Mann–Whitney U test, *p* < 0.001) and were 2 and 3 days, respectively (Appendix A). The hybrid larvae from the crossing of *I. ricinus* females and *I. persulcatus* males showed a feeding duration similar to that of the mother species (mean 2.1 ± 0.3 days) and significantly differed from that of *I. persulcatus* larvae when comparing the total data (Mann–Whitney U test, *p* < 0.001). The feeding duration of the hybrid larvae from another cross (mean 2.4 ± 0.5 days) significantly differed from that of the maternal species *I. persulcatus* (Mann–Whitney U test, *p* < 0.01) and of *I. ricinus* (Mann–Whitney U test, *p* < 0.05). At the same time, hybrids from different crosses did not significantly differ from each other in this indicator. The feeding duration of *I. ricinus* nymphs was slightly shorter than that of *I. persulcatus* (Appendix A). However, the differences in the nymphs’ feeding durations were unreliable with respect to both the original species and the hybrids. It was also not possible to reveal the dependence of the duration of feeding on the season of feeding both in nymphs and in the larvae of all groups.

When comparing the total data, the period from the engorgement to the start of molting in *I. persulcatus* larvae and hybrids obtained from crossing *I. ricinus* females and *I. persulcatus* males differed depending on the feeding season (Kruskal–Wallis H test, *p* < 0.01 and *p* < 0.05, respectively; Appendix A). These differences are not shown in Figure 3. A sharp significant increase in the period of the molting start, i.e., signs of the onset of diapause, was observed in September in *I. persulcatus* and hybrid larvae obtained from crossing *I. persulcatus* females with *I. ricinus* males (Figure 3a). In *I. ricinus* larvae and hybrids from another cross, no such sharp changes in the timing of molting were noted during the entire experimental period (March–October).

The starting times of the nymph molting of the original species differed significantly depending on the feeding season of the ticks when comparing all data groups (Kruskal–Wallis H-test, *p* < 0.01; Appendix A). These differences are not shown in Figure 3. A sharp significant increase in the period until the start of molting was observed in *I. persulcatus* nymphs feeding in October (Mann–Whitney U test, *p* < 0.01), in *I. ricinus* feeding in November (Mann–Whitney U test, *p* < 0.01), and in hybrids from the crossing of *I. persulcatus* females with *I. ricinus* males feeding in September (Mann–Whitney U test, *p* < 0.05) (Figure 3b). Interestingly, after feeding in October, the period before the start of molting was significantly longer in *I. persulcatus* nymphs than in hybrids and *I. ricinus* (Kruskal–Wallis H test, *p* < 0.01; Figure 3b). In hybrid nymphs from another cross, similar changes could not be detected due to insufficient observations.

It is worth noting that in some nymphs, we were able to record the start of ecdysis and determine their sex retrospectively. The available data are presented in Appendix A. Although males tended to molt somewhat earlier than females, the difference did not reach statistical significance even for pooled data.

Thus, the manifestation of the beginning of morphogenetic diapause in the engorged larvae and nymphs of *I. persulcatus* and hybrids from crossing *I. persulcatus* females with *I. ricinus* males was observed significantly earlier than in the larvae and nymphs of *I. ricinus* and hybrids from another cross.

## 4. Discussion

In this work, we analyzed the terms of the development of *I. ricinus*, *I. persulcatus*, and their hybrids under laboratory conditions depending on the feeding season. Since we analyzed data obtained over several years, the biological ages of the ticks (the number of days after hatching/molting) before feeding were slightly different and ranged from 30 to 90 days. We are aware that this fact may have influenced the results since it has been observed that the age of immature phases has some effect on the development of diapause [10,11] and on the duration of feeding [45]. However, the relatively small deviations and statistically significant results obtained made it possible to conclude that the analyzed data are comparable.

When obtaining hybrids, we noted that the proportion of engorged *I. persulcatus* females that mated with *I. ricinus* males was lower than not only the corresponding proportion of pure *I. persulcatus* but also the proportion of *I. ricinus* females that mated with *I. persulcatus* males. Females can attain complete nutrition only after fertilization; therefore, it is possible that *I. persulcatus* males mated with the females of another species more readily than *I. ricinus* males. In the pioneering work of Balashov et al. [20], no such relation was obtained: *I. persulcatus* and *I. ricinus* females, when crossed with a male of another species, did not differ with respect to successful feeding, weight, and oviposition success. Unfortunately, the publication did not provide data for pure species; however, the authors mention that most engorged females that mated with the males of other species started oviposition, but the number of laid eggs was significantly lower than in the parent species, and only a small number of hybrid eggs were fertile and produced viable larvae [20]. In our study, we obtained similar results: over 85% of engorged females from reciprocal crosses laid eggs, but their fertility was reduced by approximately half (Table 1). We also showed that both the females and males of hybrids from both types of reciprocal crosses were unable to reproduce themselves effectively. Out of 14 crosses involving F1 hybrids, we observed the appearance of single larvae of the second generation in only 5, and in 3 of these 5 crosses, the hybrid male mated with a female of any of the original species (Table 1).

It is known that the duration of the development of ticks and the initiation of diapause depend on the temperature, humidity, calendar feeding time, photoperiodic regime, and tick population characteristics [10,11,15,45,46]. Thus, during the development of *I. persulcatus* in laboratory conditions at a temperature of +22–25 °C, the duration of embryogenesis was 19–35 days, the metamorphosis of engorged larvae was completed in 20–28 days, and the metamorphosis of nymphs was completed in 35–43 days. When the temperature was 18–20 °C, the duration of embryogenesis increased to 30–45 days, and the molting of larvae and nymphs was observed only after 90–120 days [10,46,47]. Similar observations were made for *I. ricinus* [46,47,48]. With an increase in temperature to 28 °C, the duration of the development of both tick species was slightly reduced, with the exception of *I. persulcatus* nymphs; these nymphs developed into adults in 116 days (high temperatures suppressed their metamorphosis) [46]. Hydrothermal conditions also affect the rate of metamorphosis [10,45,48]. For example, in an experiment, *I. ricinus* larvae did not molt into nymphs and died at a relative air humidity of 75% regardless of the temperature; at 97% air humidity, the rate of larvae metamorphosis depended on the temperature: at 15 °C, less than 40% of the larvae molted in 12 weeks, while at 24 °C, all larvae molted in 6 weeks [45]. In our experiments, all ticks were kept in tubes with gradient humidity (75–100% from the top to the bottom of the tube) at room temperature (22–23 °C) under natural light conditions. These constant conditions made it possible to eliminate the effects of temperature and humidity on the development of ticks and to concentrate on the effect of the photoperiod and feeding season.

According to the obtained data, the feeding season of female ticks had an impact on the process of feeding, oogenesis, and embryogenesis, such that the females fed in the summer were characterized by the smallest weight and period from engorgement to the start of oviposition; in the spring, a shorter duration of female feeding occurred compared with the summer; and in autumn, the longest embryogenesis duration was observed (Figure 2). These patterns were observed both in pure species and in the crosses of females with the males of another species (Figure 2) and are in line with earlier observations made by other scientists. Thus, when observing the natural populations of *I. persulcatus* in Karelia, females fed in May started laying eggs 17–25 days after feeding, and those fed in June–July started laying eggs after 10–11 days depending on the temperature conditions [10]. Similar data were obtained for *I. persulcatus* in the Moscow, Kirov, and Irkutsk regions and the Krasnoyarsk Territory [15]. There were similar observations for *I. ricinus* females: in Karelia, females fed in May started laying eggs after 36–49 days, and those fed in June–July started laying eggs after 19–24 days [10]. Previously, a decisive influence of temperature on the timing of oogenesis was assumed; however, even under laboratory conditions at 22 °C, *I. ricinus* females began to lay eggs 4–7 days after feeding in May–June and 22 days or more after feeding in September–October [49].

The feeding season of immature phases also affects the duration of the development of engorged larvae and nymphs. According to the literature data, observation of the development of engorged *I. persulcatus* larvae from Karelia under laboratory conditions at a temperature of 17–23 °C (the light conditions were not specified) showed that the minimum period of molting was observed in larvae feeding in June–July (28–35 days), and a significant elongation of development was observed in September and October (90–180 days) [15]. Similar observations were made for *I. persulcatus* nymphs, wherein feeding in February–March led to molting after 2–6 months; feeding in June led to molting after 1–2 months; and feeding in September led to molting after 8 months [15]. Another author came to similar results by observing the development of *I. ricinus* in the forests of the Moscow region [50]. The period before the start of the molting of larvae and nymphs, which were fed in July, was 1–2 months shorter than that of ticks fed in May. According to our data, shorter development periods in individuals that fed in June–July were noted only for nymphs of the original species. The development of morphogenetic diapause was observed in *I. persulcatus* larvae feeding in September and in *I. persulcatus* and *I. ricinus* nymphs feeding in October and November, respectively. We failed to establish the start of diapause in *I. ricinus* larvae under laboratory conditions due to the lack of observations in November and December; however, it can be assumed that the period of diapause development in larvae is similar to that of nymphs. The period of the development of diapause in hybrids corresponded to that of the maternal species. It is worth noting that elongation of the period until the start of molting in hybrid nymphs from crossing *I. persulcatus* females and *I. ricinus* males was noted a month earlier (in September) compared with pure *I. persulcatus*, which can be explained by the lack of observations.

Thus, in this work, we have shown that under laboratory conditions that include optimal temperature and humidity, the terms of the development of *I. persulcatus*–*I. ricinus* hybrids generally correspond to those of the mother species. We are aware that the laboratory conditions that we created are far from natural. However, based on our data, we can assume that the seasonal activity of hybrid individuals in the sympatry zone of *I. persulcatus* and *I. ricinus* will be similar to that of the maternal species. To confirm this hypothesis, it is necessary to observe the natural population of ticks in the sympatric territory of two species throughout the entire period of activity.

## 5. Conclusions

Our data showed the following:−The mating of *I. ricinus* and *I. persulcatus* females with the males of another species leads to a decrease in the engorgement success of the females and a decrease in the number of hatched larvae.−In our experiments, male and female hybrids were unable to reproduce themselves effectively, even though rare F2 larvae were detected.−The duration of the feeding of *I. ricinus* and *I. persulcatus* larvae significantly differs: the larvae of hybrids from crossing *I. ricinus* females and *I. persulcatus* males have a feeding duration similar to that of the mother species, and the feeding time of the larvae of hybrids from another cross is intermediate between the values of the original species.−Under a natural daylight rhythm, the morphogenetic diapause in the larvae of *I. persulcatus* and hybrids from crossing *I. persulcatus* females and *I. ricinus* males begins in September.−Under the same experimental conditions, the morphogenetic diapause in *I. persulcatus* nymphs occurs earlier than in *I. ricinus*, in October and November, respectively.−Hybrids generally repeat the features of the life cycle of the mother species.

## Figures and Tables

**Figure 1 microorganisms-11-02252-f001:**
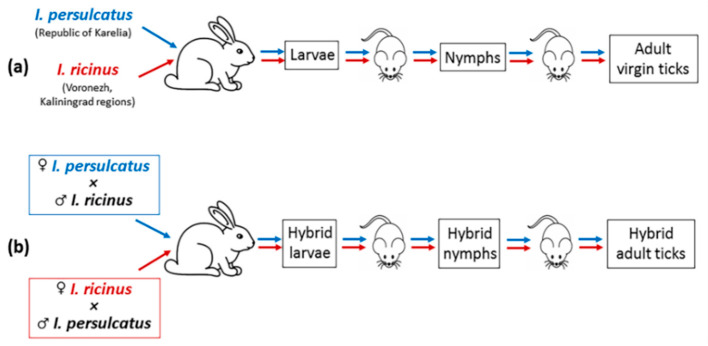
Schematic illustration of obtaining hybrids of *I. ricinus* and *I. persulcatus* ticks in the laboratory: (**a**) obtaining virgin *I. ricinus* and *I. persulcatus* females; (**b**) reciprocal crossing of the virgin females and males of other species and obtaining hybrids.

**Figure 2 microorganisms-11-02252-f002:**
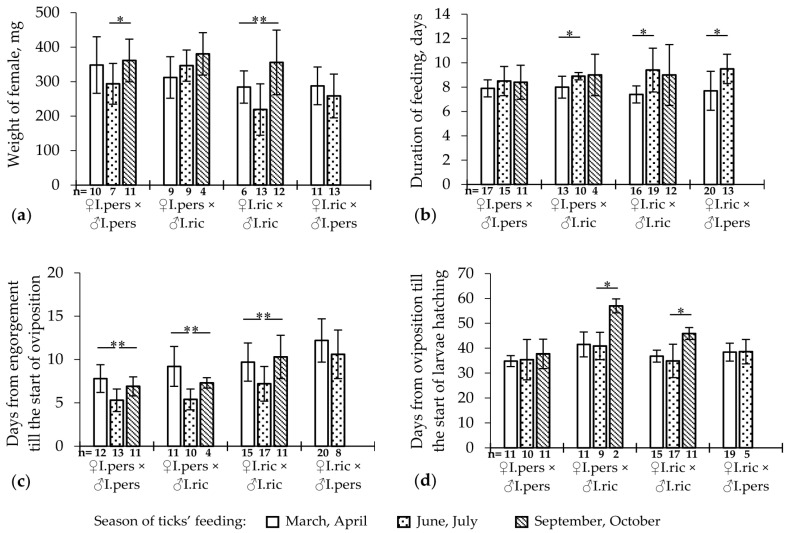
Dependence of the mean weight of engorged females (**a**), duration of feeding (**b**), oviposition of females (**c**), and hatching of the larvae (**d**) of *Ixodes ricinus* and *Ixodes persulcatus* after reciprocal crossing on the feeding season under laboratory conditions. Whiskers show standard deviation; significant differences between groups according to Mann–Whitney U test (2 groups) or Kruskal–Wallis H test (3 groups); * *p* < 0.05; ** *p* < 0.01.

**Figure 3 microorganisms-11-02252-f003:**
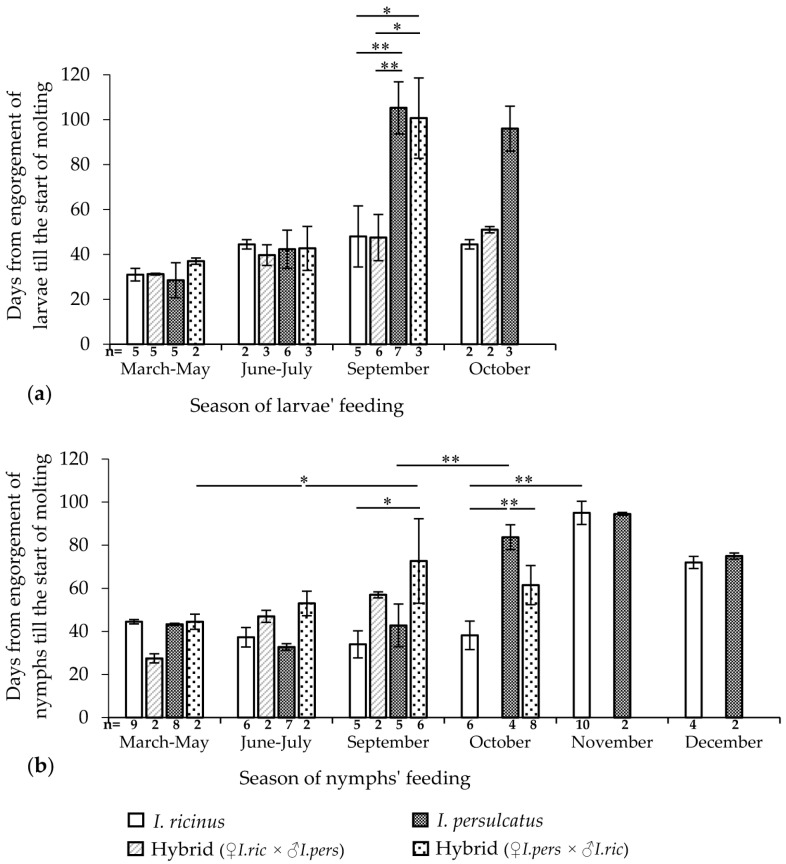
Dependence of the beginning of the molting of *Ixodes ricinus*, *Ixodes persulcatus*, and their hybrid larvae (**a**) and nymphs (**b**) on the feeding season under laboratory conditions. Whiskers show standard deviation; significant differences between groups according to the Mann–Whitney U test (2 groups) or Kruskal–Wallis H test (3 groups); * *p* < 0.05; ** *p* < 0.01.

**Table 1 microorganisms-11-02252-t001:** Characteristics of the feeding and oviposition of *I. ricinus*, *I. persulcatus*, and hybrid females depending on the species of the mating partner.

Species of Female	Species of Male	Engorgement Success of Females, Abs (%) **	Mean Weight of Engorged Females, mg ± SD	Number of Females That Laid Eggs, Abs (%)	Number of Batches Where Hatching Occurred, Abs (%)	Success of Larvae Hatching *** (High/Low)
*I. persulcatus*	*I. persulcatus*	54/73 (74.0) ^a^	339.8 ± 72.1 ^b^	43/54 (79.6)	39/43 (90.7)	High
	*I. ricinus*	25/56 (44.6) ^a,e^	310.5 ± 94.1 ^c^	23/25 (92.0)	20/23 (87.0)	Medium
	Hybrid (♀*I.pers* × ♂*I.ric*)	2/4 (50)	267.0 ± 236.2	1/2	1/1	Very low (1 LL)
*I. ricinus*	*I. ricinus*	60/82 (73.2)	284.7 ± 98.6 ^b^	51/60 (85)	50/51 (98.0) ^d^	High
	*I. persulcatus*	41/64 (64.1) ^e^	254.7 ± 81.8 ^c^	29/41 (70.7)	26/29 (89.7)	Medium
	Hybrid (♀*I.pers* × ♂*I.ric*)	4/7 (57.1)	217.0 ± 74.6	4/4 (100)	2/4 (50) ^d^	Very low (1–2 LL)
Hybrid (♀*I.pers* × ♂*I.ric*) *	*I. ricinus*	3/10 (30.0)	340.0 ± 17.4	2/3 (66.7)	0/2	-
	Hybrid (♀*I.pers* × ♂*I.ric*)	14/26 (53.9)	163.5 ± 89.9	9/14 (64.3)	2/7 (28.6)	Very low (1 LL)
Hybrid (♀*I.ric* × ♂*I.pers*)	*I. ricinus*	0/4	-	-	-	-
	*I. persulcatus*	2/4 (50)	76.5 ± 68.6	0/2	-	-
	Hybrid (♀*I.ric* × ♂*I.pers*)	1/4 (25)	35	0/1	-	-

* Parental crossing from which hybrids were obtained; ** absolute value (percentage); *** hatching success: high—more than 70%, medium—40–70%, low—10–40%, and very low—less than 10%; superscript letters indicate statistically significant differences between groups according to the Mann–Whitney U-test (weight) or Fisher’s exact test/chi-square test (*p* ˂ 0.05).

## Data Availability

The authors declare that the data supporting the findings of this study are available within the article and Appendix A and the datasets generated and analyzed for this study are available from the corresponding author upon reasonable request.

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
