# Peer review of "Development Features of Ixodes ricinus × I. persulcatus Hybrids under Laboratory Conditions"

_microorganisms, 2023, doi:10.3390/microorganisms11092252_

Round 1
Reviewer 1 Report
Belova O.A. et al.: Development features of Ixodes ricinus–Ixodes persulcatus hybrids under laboratory conditions
The authors report on crossing experiments they performed between the sheep- and the taiga ticks to extend the knowledge on hybrids potentially arising in areas where the two species’ ranges overlap. Such an experimentation is laborious and seldom undertaken, and the obtained data certainly worth publishing – conditionally on a necessary revision and completion, of course. It is laudable if an article is written concisely, however, the authors shouldn’t be laconic overly: (1) With one exception - Table 1, where the 3rd column gives some idea of the amount of material examined - sample sizes are concealed from the reader’s consideration throughout the article, leaving him/here in doubts about credibility of the authors’ claims. Note that sample means±SD - particularly if calculated from only a few observations (e.g. from 2 observations as in Tab.1, col.4, row.4!) – by no means suffice if it isn’t possible to take into account Ns as well. (2) Similarly, although it is stated that immatures were assorted by sex in a phase of engorged nymph (p.3, l.118-20), the authors make no distinction between proto-female and proto-male individuals in the statistical analysis even if there exist significant differences between them, e.g. in feeding duration (Fig. 3b). The claimed differences between nymphal pools (p.6, l.243-p.7,l.253) are thus problematical being subject to undefined F/M proportions in the samples. (3)Squeezing the information on the tested populations into the first two moments is indeed reasonable if the sample sizes are limited, however, some data are suggestive of larger Ns (unfortunately, one can only speculate..?) making it possible to assemble a sound histogram/distribution of values. Ideally, it could reveal some inheritance patterns – i.e. multimodality reflecting the split of alleles of the parental species. If adequate data is available, I encourage the authors to add histograms to the supplementary file; this could augment the scientific content of the article…
Minor issues:
Title: “Ixodes ricinus x I. persulcatus”
P.1, l.38-9 and throughout: although it is a matter of taste, I’d prefer to speak about developmental stages, and feeding phases, rather than the other way round (tick stages/instars are distinct categories whereas engorgement status is a continuous variable). Similarly, for a better orientation among multiple uses of the word “period”, I recommend to use the terms “feeding duration” and “season”, and the word “period” use when meaning a time interval in general.
P.2, l.96-7: “However, the features of the life cycle of hybrids have not yet been described, which must be taken into account when predicting the epizootic and epidemic potential of the natural foci of tick-borne infections in I. ricinus–I. persulcatus sympatry areas…” – do the authors have any evidence that the hybrids are prevalent/’contributive’ to such an extent that they can influence “the epizootic and epidemic potential of the natural foci” of TBDs ? If not, moderate this statement, pls.
P.3, l.103-4: note that it is senseless to georeference origins of the tick material with sub-metre precision – round the coordinates to a reasonable number of decimals, pls (6 decimal digits ~ 11cm, 5 digits ~ 1m, 4 digits ~ 10m, …)
P.8, l.288: “…appearance of one or two larvae...” – what an exactness! Rewrite it and be specific, pls.
P.8, l.289: “..especially when a hybrid male is crossed with a female of the original species..” – any hybrid combines in itself two parental species – which one is the “original species ”? Explain, pls.
P.8, l.308-10: “These conditions made it possible to eliminate the effect of temperature and humidity on the development of ticks and to concentrate on the effect of the photoperiod and feeding period” – this statement calls for a deeper explanation, I’m not convinced it is so…
P.9, l.364-5: it is an oxymoron. Suggestion: “in our experiments, male and female hybrids were unable to effectively reproduce themselves, even though rare F2 larvae were detected”
P.9, l.371-3: I suggest to substitute “under a natural daylight rhythm” for “under laboratory conditions”
P.9, l.374: “ Under the same experimental conditions…” ?
Table S1: insert additional columns after the 2nd, 7th, and 11th one indicating the sample sizes, pls.
Figure 1: indicate N in each column, pls.
Figure 2: ditto.
.
Author Response
The authors report on crossing experiments they performed between the sheep- and the taiga ticks to extend the knowledge on hybrids potentially arising in areas where the two species’ ranges overlap. Such an experimentation is laborious and seldom undertaken, and the obtained data certainly worth publishing – conditionally on a necessary revision and completion, of course.
Response: the authors thank the reviewer for careful reading of the manuscript and for valuable comments and suggestions. The authors believe that the manuscript has become better and clearer for the reader. All changes in the text are highlighted in yellow.
It is laudable if an article is written concisely, however, the authors shouldn’t be laconic overly:
(1) With one exception - Table 1, where the 3rd column gives some idea of the amount of material examined - sample sizes are concealed from the reader’s consideration throughout the article, leaving him/here in doubts about credibility of the authors’ claims. Note that sample means±SD - particularly if calculated from only a few observations (e.g. from 2 observations as in Tab.1, col.4, row.4!) – by no means suffice if it isn’t possible to take into account Ns as well.
Response: the authors agree with the reviewer, all figures and tables were supplemented with the sample sizes (n). In our work, we draw a conclusion only where the sample size was sufficient for statistical analysis
(2) Similarly, although it is stated that immatures were assorted by sex in a phase of engorged nymph (p.3, l.118-20), the authors make no distinction between proto-female and proto-male individuals in the statistical analysis even if there exist significant differences between them, e.g. in feeding duration (Fig. 3b). The claimed differences between nymphal pools (p.6, l.243-p.7,l.253) are thus problematical being subject to undefined F/M proportions in the samples.
Response: Available data on molting of proto-females and proto-males was added to the supplementary material (new Table S3). The sample size for proto-females and proto-males in terms of period of molting start depending on the feeding season is quite poor, and no statistically significant differences were found, even if we analyze the combined data (ignoring the feeding season). Also, the beginning of molting was recorded for a group of nymphs (about 15-20 ind), and not for individuals. In other words, the date of molting of the first nymph from the group was recorded, and it did not matter how many ticks were in the group. The number of such "male" and "female" groups within one experimental group was almost always the same. In connection with the above, the authors considered it acceptable to combine data for proto-females and proto-males in order to increase the sample size.
The text about this issue was added to the manuscript, p.7 (l 267-271)
(3)Squeezing the information on the tested populations into the first two moments is indeed reasonable if the sample sizes are limited, however, some data are suggestive of larger Ns (unfortunately, one can only speculate..?) making it possible to assemble a sound histogram/distribution of values. Ideally, it could reveal some inheritance patterns – i.e. multimodality reflecting the split of alleles of the parental species. If adequate data is available, I encourage the authors to add histograms to the supplementary file; this could augment the scientific content of the article…
Response: the authors thank for the comment. Unfortunately, in authors’ opinion, in present work the sample sizes are too small to construct meaningful diagrams of the distribution of values. The authors are planning an experiment where the characteristics of the development of individual ticks will be recorded, and in this case, the construction of such diagrams will be necessary.
Minor issues:
Title: “Ixodes ricinus x I. persulcatus”
Response: the title was corrected
P.1, l.38-9 and throughout: although it is a matter of taste, I’d prefer to speak about developmental stages, and feeding phases, rather than the other way round (tick stages/instars are distinct categories whereas engorgement status is a continuous variable). Similarly, for a better orientation among multiple uses of the word “period”, I recommend to use the terms “feeding duration” and “season”, and the word “period” use when meaning a time interval in general.
Response: the authors thank for the comment; the text was corrected
P.2, l.96-7: “However, the features of the life cycle of hybrids have not yet been described, which must be taken into account when predicting the epizootic and epidemic potential of the natural foci of tick-borne infections in I. ricinus–I. persulcatus sympatry areas…” – do the authors have any evidence that the hybrids are prevalent/’contributive’ to such an extent that they can influence “the epizootic and epidemic potential of the natural foci” of TBDs ? If not, moderate this statement, pls.
Response: the text was changed, p.3 (l 107-114)
P.3, l.103-4: note that it is senseless to georeference origins of the tick material with sub-metre precision – round the coordinates to a reasonable number of decimals, pls (6 decimal digits ~ 11cm, 5 digits ~ 1m, 4 digits ~ 10m, …)
Response: the coordinates were rounded, l 118
P.8, l.288: “…appearance of one or two larvae...” – what an exactness! Rewrite it and be specific, pls.
Response: in order to avoid complicating the discussion section, the authors have included a reference to the table 1, where the necessary details are presented
P.8, l.289: “..especially when a hybrid male is crossed with a female of the original species..” – any hybrid combines in itself two parental species – which one is the “original species ”? Explain, pls.
Response: here the authors meant any of the two original species, the text was clarified
P.8, l.308-10: “These conditions made it possible to eliminate the effect of temperature and humidity on the development of ticks and to concentrate on the effect of the photoperiod and feeding period” – this statement calls for a deeper explanation, I’m not convinced it is so…
Response: the authors meant here, that since ticks were kept under constant optimal temperature and humidity, the factors that changed and could influence on ticks development were photoperiod and feeding season.
P.9, l.364-5: it is an oxymoron. Suggestion: “in our experiments, male and female hybrids were unable to effectively reproduce themselves, even though rare F2 larvae were detected”
Response: the authors thank for the comment; the text was corrected, l 383-384
P.9, l.371-3: I suggest to substitute “under a natural daylight rhythm” for “under laboratory conditions”
Response: the authors cannot find the phrase “under a natural daylight rhythm” on line 371-373, nor in the manuscript
P.9, l.374: “ Under the same experimental conditions…” ?
Response: the authors cannot find the phrase “Under the same experimental conditions…” on line 374, nor in the manuscript
Table S1: insert additional columns after the 2nd, 7th, and 11th one indicating the sample sizes, pls.
Response: the sample sizes were added to the Table S1 and S2
Figure 1: indicate N in each column, pls.
Figure 2: ditto.
Response: the sample sizes were indicated in each column of the figures 2 and 3.
Reviewer 2 Report
Ticks Ixodidae are an interesting group of parasitic mites, widely studied in the context of pathogen transmission. However, less research is concerned with their biodiversity, variability, biology, life cycles, and complicated parasite-host relationships. While without these data, it is but yet difficult to understand their function as vectors and the complex mechanisms of disease transmission. In this context, the manuscript brings new and interesting data.
In my opinion, however, the manuscript should be slightly revised, and refined.
DETAILED COMMENTS
The research problem is clearly formulated and presented in the introduction; however, the goals and meaning of the research are less clear.
Besides might be worth adding other literature data, beside the work of Russian researchers (e.g. information on the ranges of occurrence of ticks, data indicating that probable hybrids - individuals with intermediate features - occur in areas where both tick species coexist). Especially since citing literature is selective, many works of Russian authors were quoted, publications of other European researchers were not taken into account. The manuscript was submitted to an international journal, so representative papers should be included.
Data on seasonality and seasonal activity of ticks are also incomplete;
according to the authors, the life cycles of these ticks are well known. Meanwhile, as a result of climate change (warming), the periods of occurrence (and feeding) of individual tick life stages are changing.
Discussion - extensive, includes an analysis of many detailed data - many different references are quoted to the development of life stages under different environmental conditions, but it is not always clear what conclusions this leads to, what results from them. The discussion should be put in order.
The authors mention that the tested ticks are important as vectors of pathogens; maybe (taking into account the subject of the journal) it would be good to consider whether the conclusions of the research may be relevant in the context of tick parasitism and pathogen transmission.
Lines 27, 108 - In the scientific journal the scientific nomenclature should be precise. When first entering each scientific name of the species - full name (with author's name and description date), and systematic position.
Lines 27-35 Citations concerning the range of occurrence of the studied ticks concern mainly Ixodes ricinus (and these are partly randomly selected items); no original citations confirming the data on the distribution range of I. persulcatus.
Imprecise terminology, e.g.line 101 – “Ticks and animals” - ticks also belong to the animal kingdom; this was probably not about animals, but about tick hosts.
Line 108 - Oryctolagus cuniculus - italic
The manuscript should be reviewed by a native speaker.
Sometimes unusual sentence structure (as when using an online language translator) e.g. "obtaining virgin I. ricinus and I. persulcatus females".
Author Response
Ticks Ixodidae are an interesting group of parasitic mites, widely studied in the context of pathogen transmission. However, less research is concerned with their biodiversity, variability, biology, life cycles, and complicated parasite-host relationships. While without these data, it is but yet difficult to understand their function as vectors and the complex mechanisms of disease transmission. In this context, the manuscript brings new and interesting data.
In my opinion, however, the manuscript should be slightly revised, and refined.
Response: the authors thank the reviewer for careful reading of the manuscript and for valuable comments and suggestions. The authors tried to take into account all the comments of the reviewer. All changes in the text are highlighted in yellow. The English language was corrected by the MDPI English editing service.
DETAILED COMMENTS
The research problem is clearly formulated and presented in the introduction; however, the goals and meaning of the research are less clear.
Response: the authors tried to formulate the goals of the research more clear, p.3 (l 101-114)
Besides might be worth adding other literature data, beside the work of Russian researchers (e.g. information on the ranges of occurrence of ticks, data indicating that probable hybrids - individuals with intermediate features - occur in areas where both tick species coexist). Especially since citing literature is selective, many works of Russian authors were quoted, publications of other European researchers were not taken into account. The manuscript was submitted to an international journal, so representative papers should be included.
Response: the authors thank for the comment and agree with the reviewer. Some text and references were added, p.1,2 (l 30-33, 99-100)
Data on seasonality and seasonal activity of ticks are also incomplete;
according to the authors, the life cycles of these ticks are well known. Meanwhile, as a result of climate change (warming), the periods of occurrence (and feeding) of individual tick life stages are changing.
Response: The authors agree with the reviewer that the description of seasonal tick activity in the manuscript is incomplete. The introduction describes the general characteristics of the seasonality of I. ricinus and I. persulcatus ticks in temperate climate. However, due to the vast range and adaptation of ticks to different climatic conditions, the seasonal tick activity varies and has its own characteristics not only in different species, but also in different populations of ticks. The authors fear that a more detailed description of these features will lead to the transformation of an experimental article into a review.
Discussion - extensive, includes an analysis of many detailed data - many different references are quoted to the development of life stages under different environmental conditions, but it is not always clear what conclusions this leads to, what results from them. The discussion should be put in order.
Response: The Discussion is written in such a way that the order in which the results are discussed corresponds to the arrangement of the conclusions in the Conclusion section. The authors would be grateful to the reviewer if he points out the exact places in the discussion that are not clear and might confuse the reader.
The authors mention that the tested ticks are important as vectors of pathogens; maybe (taking into account the subject of the journal) it would be good to consider whether the conclusions of the research may be relevant in the context of tick parasitism and pathogen transmission.
Response: the text about pathogen prevalence in allopatric and sympatric zones was added to the introduction section, p.2 (l 80-91)
Lines 27, 108 - In the scientific journal the scientific nomenclature should be precise. When first entering each scientific name of the species - full name (with author's name and description date), and systematic position.
Response: the species names were corrected, l 27-28, 123, 126
Lines 27-35 Citations concerning the range of occurrence of the studied ticks concern mainly Ixodes ricinus (and these are partly randomly selected items); no original citations confirming the data on the distribution range of I. persulcatus.
Response: the references were added, l 32
Imprecise terminology, e.g.line 101 – “Ticks and animals” - ticks also belong to the animal kingdom; this was probably not about animals, but about tick hosts.
Response: the name of the subsection was corrected, l 116
Line 108 - Oryctolagus cuniculus - italic
Response: corrected, l 123
Round 2
Reviewer 1 Report
Belova O.A. et al.: Development features of Ixodes ricinus x Ixodes persulcatus hybrids under laboratory conditions. V2
I acknowledge that the manuscript has been amended to conform to my suggestions. I recommend a minor revision (a thorough linguistic revision is understood..).
P.1, l. 27: for the sake of brevity, “Linnaeus” could be substituted with “L.”
P.2, l.85-6: put the citations in square brackets at the end of that statement (after “sympatry”), pls
P.8, l.267-71: this paragraph isn’t clear and should be re-phrased (suggestions: “..in some cases, engorged nymphs were assorted by sex, and we were able to observe the start of molting of proto-female and proto-male nymphs..” -> “..in some nymphs we were able to spot the start of ecdysis and determine their sex retrospectively..”; “Despite some trend of the earlier molting start of males, observed differences are statistically insignificant even when merged data is analysed” -> „Although males tended to moult somewhat earlier than females, the difference didn‘t reach statistical significance even for data pooled across all experiments“).
P.9, l.306-7: “.. the appearance of occasional single larvae of the second generation is possible..” – this is a formulation worthy of an attorney, not a scientist! You should unequivocally state whether you detected several few cases/ a single case/ or speculate.
Ad Title: “Ixodes ricinus x I. persulcatus”
Response: the title was corrected
Double check it !
Ad P.9, l.371-3: I suggest to substitute “under a natural daylight rhythm” for “under laboratory conditions”
Response: the authors cannot find the phrase “under a natural daylight rhythm” on line 371-373, nor in the manuscript
Неудивительно, потому что я предложил вам заменить выраже́нием “under athe natural daylight rhythm” существующее выраже́ние “under laboratory conditions”, не наоборот…
Ad P.9, l.374: ditto
Author Response
I acknowledge that the manuscript has been amended to conform to my suggestions. I recommend a minor revision (a thorough linguistic revision is understood..).
P.1, l. 27: for the sake of brevity, “Linnaeus” could be substituted with “L.”
Response: the text was corrected, l 27
P.2, l.85-6: put the citations in square brackets at the end of that statement (after “sympatry”), pls
Response: the text was corrected, l 86
P.8, l.267-71: this paragraph isn’t clear and should be re-phrased (suggestions: “..in some cases, engorged nymphs were assorted by sex, and we were able to observe the start of molting of proto-female and proto-male nymphs..” -> “..in some nymphs we were able to spot the start of ecdysis and determine their sex retrospectively..”; “Despite some trend of the earlier molting start of males, observed differences are statistically insignificant even when merged data is analysed” -> „Although males tended to moult somewhat earlier than females, the difference didn‘t reach statistical significance even for data pooled across all experiments“).
Response: the authors thank the reviewer for the suggestions; the text was corrected, l 267-270
P.9, l.306-7: “.. the appearance of occasional single larvae of the second generation is possible..” – this is a formulation worthy of an attorney, not a scientist! You should unequivocally state whether you detected several few cases/ a single case/ or speculate.
Response: the text was rephrased, l 305-308
Ad Title: “Ixodes ricinus x I. persulcatus”
Response: the title was corrected
Double check it !
Response: the title was checked. The authors can change the title only in the word-file of the manuscript. The authors, unfortunately, cannot change the title that was entered into the manuscript submission system. The authors will check the title again at the proofread stage
Ad P.9, l.371-3: I suggest to substitute “under a natural daylight rhythm” for “under laboratory conditions”
Response: the authors cannot find the phrase “under a natural daylight rhythm” on line 371-373, nor in the manuscript
Неудивительно, потому что я предложил вам заменить выраже́нием “under athe natural daylight rhythm” существующее выраже́ние “under laboratory conditions”, не наоборот…
Ad P.9, l.374: ditto
Response: the authors apologize for the misunderstanding; the text was corrected, l 390, 393